# ROBUST QUANTUM NEURAL NETWORKS AGAINST DYNAMIC NOISE LANDSCAPE IN THE NISQ ERA

## ABSTRACT

Quantum machine learning, an emerging field in the noisy intermediate-scale quantum (NISQ) era, faces significant challenges in error mitigation during training and inference stages. Current noise-aware training (NAT) methods typically assume static error rates in quantum neural networks (QNNs), often neglecting the inherently dynamic nature of such noise. By addressing this oversight, our work recognizes the dynamics of noise in the NISQ era, evidenced by fluctuating error rates across different times and qubits. Moreover, QNN performance can vary markedly depending on the specific locations of errors, even under similar error rates. This variability underscores the limitations of static NAT strategies in addressing the dynamic nature of noisy environments. We propose a novel NAT strategy that adapts to both standard and fatal error conditions, cooperating with a low-complexity search strategy to efficiently locate fatal errors during optimization. Our approach marks a significant advancement over current NAT methods by maintaining robust performance in fatal error scenarios. Evaluations validate the efficacy of our strategy against fatal errors, while maintaining performance comparable to state-of-the-art NAT approaches under various error rates.

## 1 INTRODUCTION

Quantum machine learning (QML) has found great advancement along with modern classical ML in the areas of chemistry Sajjan et al. (2022), physics Guan et al. (2021), natural language processing Guarasci et al. (2022), etc. Specifically, the variational quantum algorithm (VQA) Cerezo et al. (2021) is recognized as one of the most promising solutions to near-term QML tasks. The VQA is performed in the hybrid classical-quantum optimization scenario, where parameterized quantum circuits (PQCs), a.k.a. *ansatz*, produce measurements of quantum states on the quantum computer, and the classical computer performs parameter optimization of PQCs through gradient-based methods Simeone et al. (2022). We demonstrate this procedure in Figure 1, taking qautnum neural

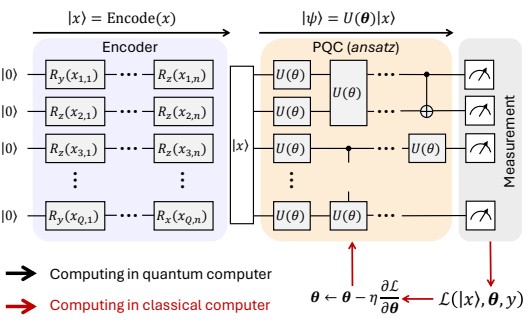

Figure 1: A demonstration of QNN task. Input $x$ is encoded as a state vector $|x\rangle$ using rotation gates Wang et al. (2022a), which is fed into PQC to prepare a final state vector $|\psi\rangle$ for measurement.

network (QNN) as an example, and further details of PQC optimization are provided in Appendix A.

Different from classical circuits, the execution on PQCs faces a more rigorous situation with noise. As quantum computers move to the noisy intermediate-scale quantum (NISQ) era, errors in qubits and quantum operations are inevitable due to the dynamic nature of the quantum system and its environment Preskill (2018). Under the complexity of quantum systems and the fragility of qubits, basis gates usually have error rates on the scale of $10^{-4}$ to $10^{-2}$, as reported in IBM quantum calibration data, while classical logic gates already achieve error rates lower than $10^{-6}$ decade ago Mielke et al. (2008). Thus, error mitigation in PQCs is critical for robust execution on quantum computers, where mitigation strategies have been proposed during QNN training Wang et al. (2022b) or execution Cai et al. (2023); Endo et al. (2021); Strikis et al. (2021).

In this work, we focus on noise-aware training (NAT) for QNN models, aiming to enhance the robustness of PQC against errors. The current NAT approach proposed by Wang et al. (2022b) involves deliberately introducing simulated noise into quantum gates, based on a predefined error rate. This technique is designed to improve the robustness of PQC to a specific error rate. Despite the pronounced effectiveness, we highlight a critical limitation in the current NAT strategy and PQC simulators, which rely on static error rates during noise analysis. ***In reality, the noise landscape is dynamic across both time and qubits.*** Recent studies have demonstrated that noise levels in quantum computers can fluctuate significantly over time, even within a transient time window of operation Ravi et al. (2023). In our work, we also emphasize the dynamic nature of noise across different qubits. In extreme cases, the error rate for a quantum gate can vary dramatically — from as high as 8% on one qubit to as low as 0.1% on another — illustrated in Figure 2. Such a highly variable noise landscape renders the static error rate assumption in NAT insufficient for practical applications. For instance, a qubit in an optimized QNN model that was assumed to have a low error rate during simulation could be mapped to a physical qubit with a significantly higher error rate or might experience increased errors over extended runtime.

Additionally, current approaches to evaluating errors in noisy QNN models predominantly focus on the probability of errors, rather than assessing the actual detrimental impact these errors have on QNN performance. Practically, even errors with identical probabilities can have vastly different consequences. Certain errors may have a negligible influence on the QNN's inference performance, while some fatal errors can degrade its performance to levels worse than random guessing. The situation is more critical when qubits with possible fatal errors are mapped to high-error-rate physical qubits, posing a significant risk to the robustness of QNN inference.

In this paper, we characterize the influence of dynamic noise on PQCs, and demonstrate the scenarios where state-of-the-art (SOTA) NAT strategy is not competent. While it can mitigate errors occurring at predictable rates, it falls short in managing fatal errors that critically undermine QNN performance. Our analysis underscores the need for a more comprehensive NAT strategy that transcends static error rate consideration and effectively mitigates fatal errors. To tackle this challenge, we propose a low-complexity search strategy. It can search for fatal errors along with QNN training, and evolve with PQC parameters simultaneously. Rather than randomly sampling errors, we utilize the searched fatal errors to boost the robust QNN training. We highlight our contributions as follows:

- We reveal the significant influence of dynamic noise landscape on QNN models. The uncertainty of error rates hinders the practice of current NAT strategies on static error rates.

- We quantify the negative influence of error cases on QNN models, and propose the analysis methods for the fatal errors on a QNN model.

- To the best of our knowledge, this is the first NAT method that is independent of error rates, but concentrates on fatal errors of PQCs. Our strategy aims to develop QNN models adaptive to various noise distributions and with good tolerance to these fatal error cases.

- We thoroughly evaluate the effectiveness of our strategy, which greatly improves the robustness of QNN models against fatal errors. Also, our strategy can achieve performance comparable to that of the SOTA NAT strategy under various error rates.

## 2 PRELIMINARY

### 2.1 DEFINITION

We follow the QNN model summarized in Mitarai et al. (2018), composed of an input encoder, a PQC, and a measurement module. The PQC is optimized for a set of training data and applied for further inference on quantum computers.

**Definition 2.1** (QNN feed-forward). A QNN model starts with a predefined encoder Yan et al. (2016); Wang et al. (2022a;b) to prepare the query sample $x$ as a quantum state vector $|x\rangle$. It then applies a sequence of parameterized unitary gates $U_g(\theta_g) \in \mathcal{G}$ represented by unitary matrix $U(\boldsymbol{\theta}) = \prod_{g=G}^{1} U_g(\theta_g)$. The final quantum state is denoted as $|\psi\rangle = U(\boldsymbol{\theta})|x\rangle$. Then, the measurement of PQC is tackled by evaluating an observable $B$ on $|\psi\rangle$, which is expressed as

$$f(|x\rangle, \boldsymbol{\theta}) = \langle\psi| B |\psi\rangle = \langle x| U^\dagger(\boldsymbol{\theta}) B U(\boldsymbol{\theta}) |x\rangle \tag{1}$$

In the following, we use $f(|x\rangle, \boldsymbol{\theta})$ to denote the expected measurement results of PQC.

**Definition 2.2** (Noise model). To simulate noise effects in PQCs, we consider bit flip errors (`Pauli-X`), phase flip errors (`Pauli-Z`), and bit-phase flip errors (`Pauli-Y`), as outlined in Wang et al. (2022b). Other common errors, such as decoherence, can be mitigated without additional resource overhead during execution Smith et al. (2022); yet, mitigating Pauli errors typically requires extra qubits and operations Terhal (2015); Acharya et al. (2023); Chen et al. (2021). Therefore, developing QNNs with inherent resilience to Pauli errors during training can significantly alleviate extensive error correction during execution, enhancing their performance on quantum computers. Additionally, post-training error mitigation methods, which are independent of our approach, can complement our in-training strategy. This combination is further explored in Appendix B. We introduce two error modeling methods Nielsen & Chuang (2010) involved in our analysis:

**Probabilistic Modeling.** Denoting the density matrix of a pure state $|\phi\rangle$ as $\rho = |\phi\rangle \langle\phi|$, the error channel of error $E$ with probability $p$ is represented by the mixed state as $\mathcal{E}(\rho) = (1-p)\rho + pE\rho E$. The measurement of PQC is expressed as

$$f(|x\rangle, \boldsymbol{\theta}) = \mathbb{E}_{\mathcal{E} \sim \mathcal{P}(p^X, p^Y, p^Z)} \text{Tr}[B\rho_G], \text{ where } \rho_G = \left(\prod_{g=1}^{G} \mathcal{E}_g \circ U_g(\theta_g)\right)(\rho_0) \qquad (2)$$

The noisy operation on the $g$-th gate conducts $\mathcal{E}_g \circ U_g(\theta_g)(\rho) = \bar{p}_g I \rho' I + p_g^X X \rho' X + p_g^Y Y \rho' Y + p_g^Z Z \rho' Z$, where $\rho' = U_g(\theta_g)\rho U_g^\dagger(\theta_g)$, $\bar{p}_g = 1 - p_g^X - p_g^Y - p_g^Z$, and $\rho_0 = |x\rangle \langle x|$. The error of each gate $\mathcal{E}_g \in \mathcal{E}$ follows the distribution of $\mathcal{P}_g = (\bar{p}_g, p_g^X, p_g^Y, p_g^Z)$ representing the probability of error-free, `Pauli-X`, `Pauli-Y`, and `Pauli-Z` error on $g$-th gate. This probabilistic noise model estimates the expected PQC measurement given the predefined error rates of each quantum gate.

**Deterministic Modeling.** Rather than a general expectation, the deterministic noise model estimates the PQC output in certain error cases. Assuming an error $E_g$ (as an operation) could occur on the gate $U_g(\theta_g)$, e.g., $U_g(\theta_g|E_g) = E_g U_g(\theta_g)$, we define the measurement of PQC as

$$f(|x\rangle, \boldsymbol{\theta}|\boldsymbol{E}) = \langle\psi| B |\psi\rangle, \text{ where } |\psi\rangle = U(\boldsymbol{\theta}|\boldsymbol{E}) |x\rangle \qquad (3)$$

The noisy PQC conducts $U(\boldsymbol{\theta}|\boldsymbol{E}) = \prod_{g=G}^{1} E_g U_g(\theta_g)$ on $|x\rangle$, where $\boldsymbol{E} \in \{I, X, Y, Z\}^G$ is a specific error case. Deterministic modeling tackles the analysis of error effect under certain errors, such as systemetic errors Khodjasteh & Viola (2009) and coherent errors Greenbaum & Dutton (2017). In our work, we utilize this model to evaluate the worst-case errors for PQC measurement.

## 2.2 Scope of Our Work

**Application Scenario.** We aim to mitigate the dynamic error effect on quantum computers by training a robust model for *future use*. A robust QML model should perform well under dynamic and unpredictable noise. Therefore, QNN models pre-trained through classical simulations or classical-quantum hybrid algorithms are ideal for this scenario. Other VQA tasks, such as VQE Tilly et al. (2022), fall outside our scope but benefit from noise mitigation techniques like quantum error correction codes Acharya et al. (2023). For instance, once the state is optimized and prepared in VQE, no further execution of the PQC is needed[1]. In such tasks, real-time error mitigation methods, including surface codes Acharya et al. (2023) and iterative skipping strategies Ravi et al. (2023), are more appropriate. Related work is provided in Appendix C.

**Case Study.** In the following analysis, validation is provided as a numerical proof of concept along with the analytical dicussion. Without loss of generality, we take a case study from a 4-qubit QNN, whose PQC contains 3 layers. Each layer is composed of 4 `U3` gates and 4 `CU3` gates in a cyclical manner Wang et al. (2022a). This model is trained for an MNIST-2 task that classifies digits 3 and 6.

## 3 Dynamic Noise and Fatal Error on QNN

### 3.1 Severe Error Variation on Time Scale and Qubit Scale

In previous NAT on QNN, the model parameters are usually optimized under the static noise assumption. However, transient errors have recently been recognized for both long-term and short-

---

[1]This excludes extreme cases, such as reusing or analyzing the found state in the future.

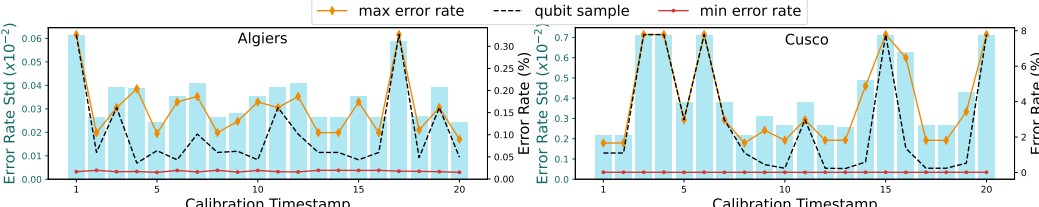

Figure 2: The demonstration of large variation of qubit error rates on both time scale and qubit scale. We record the `Pauli-X` gate error rates of all qubits of two IBM quantum computers, `ibmq_algiers` (27 qubits) and `ibmq_cusco` (127 qubits). We sample 20 calibration data in a consecutive 72-hour frame. The barplot (left y-axis) demonstrates the standard deviation of error rates on qubits of each calibration, along the time scale. The line plot (right y-aixs) shows the maximum/minimium qubit error rate of all qubits for each calibration, and the dash line highlights the qubit error rate track which has the largest variation along the calibration time window. Calibration data on other quantum computers are provided in Appendix I.1.

term quantum operations Ravi et al. (2023), where the characteristic of qubits (e.g., gate error rate) could vary significantly along the ***time*** scale.

Even worse, large variations of qubit characteristics also exist along the ***qubit*** scale. In Figure 2, we demonstrate the dynamic noise landscape from two IBM commercial quantum computers, on the time scale and the qubit scale. During a long-term run, the gate error rate on a certain qubit (e.g., shown as black dash lines in Figure 2) could vary by an order of magnitude in the worst case. On the qubit scale, the error rates of different qubits also have a significant variation. In some extreme cases, the error rate on one qubit can be 80 times greater than that on another. Thus, for a well-trained QNN, different mappings from its qubits to the physical qubits on quantum computers can induce significant error rate divergence on each qubit. Overall, the transient error rates on qubits are quick-changing and unpredictable, making it difficult to model them and provide a reliable prediction.

**Error Rate Variation Analysis.** We consider a simplified example of error rate deviation in probabilistic modeling (Eq. 2). Specifically, we assume a small deviation $\Delta p$ in the `Pauli-X` error rate at the $g$-th gate, such that $(p_g^X)' = p_g^X + \Delta p$ and $(\bar{p}_g)' = \bar{p}_g - \Delta p$. We evaluate the impact of this error rate deviation on the PQC measurement, $\Delta f = f(|x\rangle, \boldsymbol{\theta}|p_g^X + \Delta p) - f(|x\rangle, \boldsymbol{\theta}|p_g^X)$. Since $f(|x\rangle, \boldsymbol{\theta})$ is smooth, we apply a Taylor expansion around $p_g^X$ and estimate the measurement deviation as $\Delta f \approx \partial f / \partial p_g^X \cdot \Delta p$. The resulting measurement influence is derived as:

$$\Delta f = \text{Tr}[\rho_{g+}(X\rho'_{g-1}X - \rho'_{g-1})]\Delta p, \text{ where } \rho'_{g-1} = U_g(\theta_g)\left[\left(\prod_{i=1}^{g-1}\mathcal{E}_i \circ U_i(\theta_i)\right)(\rho_0)\right]U_g^\dagger(\theta_g)$$

$\rho_{g+}$ is the back-propagation operator from the observer $B$. Detailed derivation is provided in Appendix D. In the worst case, where $X\rho'_{g-1}X = -\rho'_{g-1}$, the measurement is significantly affected, resulting in $\Delta f = \text{Tr}[-2\rho'_{g-1}\rho_{g+}]\Delta p$, even with a small error rate deviation. Extending this analysis to larger error rate deviations and potentially occurring across multiple qubits, probabilistic modeling may lead to significant deviations in measurement predictions. This indicates that static noise analysis, which assumes fixed error rates, is inadequate for capturing realistic dynamic noise across both time and qubit scales. Consequently, current NAT strategies and evaluation methods based on static error rates cannot effectively train or evaluate QNNs under dynamic noise in practice.

## 3.2 FATAL ERRORS POISONING QNN ACCURACY

Besides the error rate variation, not all gates in a PQC have the same sensitivity to errors. Although this is a straightforward phenomenon, the influence of gate error on QNN performance is astonishing. We present this assessment of QNN in Figure 3, under a minimal error scenario where only one single gate is affected by error. The result is intriguing, since even one gate error can significantly compromise the QNN model into a useless level. Specifically, there are 9 out of 72 errors, for a QNN trained without noise consideration, that result in performance below random guess (0.5 for the MNIST-2 task). Even with SOTA noise injection training Wang et al. (2022b),

there are still 4 errors that lead to worse than random guess. This underscores that errors at different locations can cause widely disparate levels of performance degradation, even with similar error rates. On real quantum computers, the occurrence of such a fatal error can render the QNN inference utterly unreliable. Under the dynamic noise, the coming of fatal error cases is also unpredictable.

As a preview, our strategy significantly enhances QNN performance across all error cases, ensuring that predictions consistently exceed random guesses in all scenarios explored in our case study. Furthermore, accuracy histograms indicate that SOTA noise injection training generally improves model accuracy under most error conditions over the error-free training (baseline). However, this approach falls short in addressing specific low-accuracy, or fatal, error scenarios effectively. In contrast, our method successfully mitigates the impact of these fatal errors without compromising model performance in other error scenarios.

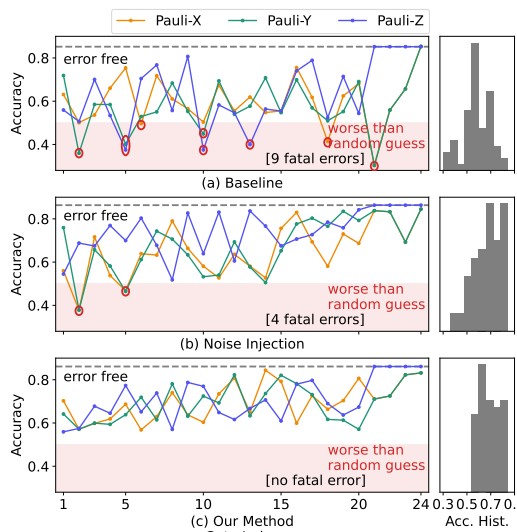

Figure 3: The inference accuracy of QNN which is trained by (a) no error consideration (b) noise-injection strategy and (c) our method (Section 4), for the MNIST-2 task, assuming only one gate occurring logical error. Right plots are the corresponding histograms of accuracies. The PQC in QNN has 24 gates, where each gate could have error from {Pauli-X, Pauli-Y, Pauli-Z}.

**How does the fatal error analysis behave as a complement to noisy accuracy evaluation?** The most commonly used metric to evaluate QNN performance is the inference accuracy at specific error rates, which we will refer to as *Noisy Accuracy*. This metric determines QNN performance by introducing errors at predefined rates across individual samples or batches, where the negative impact of fatal errors is diluted over multiple samples. Consequently, this approach evaluates only the average performance of a noisy QNN, not its variation. Besides, assessing accuracy at fixed error rates does not accurately reflect the real-time performance of a QNN on a quantum computer due to the dynamic nature of quantum noise. A more reliable measure of QNN robustness in noisy conditions is the performance under fatal error scenarios, which we call *Fatal Accuracy*. This metric, focusing on QNN behavior under worst-case conditions, provides a critical measure of robustness by evaluating variation in performance across different noisy environments. A QNN model that scores high in both noisy and fatal accuracies demonstrates not only strong average predictive performance under noise but also consistent robustness across various noisy conditions.

## 3.3 WORST CASE ANALYSIS

Evaluating all possible error scenarios during QNN training is both time-consuming and impractical, especially since many of these scenarios have negligible probabilities. Instead, it is more meaningful to focus on error cases with relatively high probabilities, as these are the most common during quantum execution. For example, with an average error rate of 0.001 (such as in ibmq_cusco) across a 100-gate PQC, the conditional probability that one or two errors occur given the error occurring, is $P(N_{\mathcal{E}} \leq 2 | N_{\mathcal{E}} \geq 1) \approx 99.84\%$. This indicates that nearly all error events involve one or two gate errors. Therefore, we emphasize concentrating on these relatively "high-probability" errors, e.g., considering up-to-two gate errors (denoted as $M = 2$), since they are the most likely to affect the model's performance when errors occur. The specific collection strategy of these error cases is discussed in Appendix E. In our analysis, we locate fatal errors in the high-probability error set and utilize the deterministic noise model described in Eq. 3.

**Definition 3.1** (Fatal loss $\mathcal{L}_{sup}$) Given an error set $\Omega_{\boldsymbol{e}}$ and input-label pair $(x, y)$, we define the fatal loss to the PQC measurement $f(|x\rangle, \boldsymbol{\theta})$, as the supremum of losses caused by all $\boldsymbol{E} \in \Omega_{\boldsymbol{e}}$,

$$\mathcal{L}_{sup}(\Omega_{\boldsymbol{e}}, x, y) = \sup_{\boldsymbol{E} \in \Omega_{\boldsymbol{e}}} \{\mathcal{L}(f(|x\rangle, \boldsymbol{\theta}|\boldsymbol{E}), y)\} \tag{4}$$

Accordingly, the QNN fatal accuracy evaluates the worst-case performance of PQC measurement against all noise cases in $\Omega_e$. That is, the "worst-case" performance refers to the maximum loss that surpasses other losses with high probability. Our metric in Eq. 4 does not involve gate error rates during evaluation; thus, the evaluation remains fair and robust against dynamic noise changes in quantum computers. Please note that although we consider the *general/average* error rate when collecting $\Omega_e$ to narrow the scope of noise events under consideration, this approach does not affect the dynamic noise landscape or compromise the error-rate independence of our fatal loss metric. This is because large fluctuations in few qubits' error rates can be effectively averaged out in the general error rate without introducing significant bias; meanwhile, a simultaneous and sudden change in all qubits' error rates is improbable during normal operation of a quantum computer.

To conclude, there are two challenges for noise-aware training on QNN models: ① **noisy accuracy challenge**—Since there is no obvious routine for the error rate change on qubits, it is necessary to develop a general NAT strategy, so that the optimized PQC is supposed to perform well under various error rates on average. ② **fatal accuracy challenge**—The optimized PQC should have acceptable performance even with high-probability fatal errors. A desired NAT strategy should be error-rate independent, i.e., equitable to all error rates without deliberate bias, so that the performance loss of QNN under a dynamic noise landscape can be alleviated.

## 4 METHODOLOGY: EQUITABLE TRAINING WITH LOW-COMPLEXITY SEARCH

### 4.1 PROBLEM FORMULATION

We aim to co-optimize the noisy accuracy of QNN models and the fatal accuracy given the high-probability error set. While current NAT strategies optimize robust QNN models by randomly sampling errors from all possible error cases, our approach specifically considers fatal accuracy by actively searching for and sampling the fatal errors during optimization. In this error-rate-independent optimization problem, we do not rely on specific error rates; therefore, we treat the first component as an error-free term. The optimization problem is formulated as follows:

$$\min_{\boldsymbol{\theta}} \quad \mathcal{L}\left(f(|\mathbf{X}\rangle, \boldsymbol{\theta}), \mathbf{Y}\right) + \lambda\mathcal{L}_{sup}(\Omega_{\boldsymbol{e}})$$
$$\text{s.t.} \quad \mathcal{L}_{sup}(\Omega_{\boldsymbol{e}}, \mathbf{X}, \mathbf{Y}) = \mathbb{E}_{(\mathbf{X},\mathbf{Y})\sim\mathcal{X}\times\mathcal{Y}} \sup_{\boldsymbol{E}\in\Omega_{\boldsymbol{e}}} \left\{\mathcal{L}\left(f(|\mathbf{X}\rangle, \boldsymbol{\theta}|\boldsymbol{E}), \mathbf{Y}\right)\right\} \tag{5}$$

where $(\mathbf{X}, \mathbf{Y})$ is the input-label pairs following distribution in sample space $\mathcal{X} \times \mathcal{Y}$, and $\lambda > 0$ is a hyperparameter to balance the focus on error-free accuracy and fatal accuracy.

### 4.2 LOW-COMPLEXITY SEARCH

To determine the supremum of losses caused by high-probability error cases, the most straightforward method is to exhaustively traverse all error cases in $\Omega_e$. However, the cardinality of the error set $|\Omega_e|$ actually increases polynomially along the PQC scale, i.e., the gate number $G$. Since $\mathcal{L}_{sup}$ must be evaluated for each data batch and at every iteration during training, the computational time quickly becomes impractical as the PQC scales up. Therefore, it is essential to adopt low-complexity yet effective search strategies to address this scalability challenge.

**Scalable Evolution-Based Low-Complexity Search.** We adopt the idea from genetic algorithm Mirjalili & Mirjalili (2019) to search for fatal errors during each training iteration. To balance the search effort and effectiveness, we propose allowing the error cases to evolve alongside the model optimization. More design details and comparisons with other search strategies are provided in Appendix F. Under this strategy, the search complexity in each iteration only depends on the error population $N_{pop}$ in current generation and the number of data batches.

**Remark: How does our low-complexity search perform in estimating the fatal loss?** The evaluation of $\mathcal{L}_{sup}(\Omega_e)$ must be performed repeatedly during QNN training, thus the key to reducing time consumption is to use fewer samples to derive a highly accurate estimate. In Table 1, we demonstrate the search procedure between brute-force search, random search, a sequential search strategy, and our low-complexity search. We construct three QNNs with different scales ($G = 24, 40, 56$) on the MNIST-2 task, and vary $\Omega_e$ by collecting all cases up to two ($M = 2$) and three ($M = 3$) errors. While brute-force search explores all error cases in $\Omega_e$, it guarantees an accurate

Table 1: Estimated $\mathcal{L}_{sup}(\Omega_e)$ on a data batch and time consumption on different search strategies. The first column is the QNN scale and the maximal expected gate errors in $\Omega_e$, i.e., $(G, M)$. The sequential search is developed from greedy algorithm, which is lightweight but not scalable (see Appendix F.1). The search time is measured on Intel i7 @3.80GHz CPU.

| Search Size | BruteForce | | RandomSearch | | SequentialSearch | | **ours** | |
|---|---|---|---|---|---|---|---|---|
| | $\mathcal{L}_{sup}(\Omega_e)$ | Time(s) | $\widehat{\mathcal{L}}_{sup}(\Omega_e)$ | Time(s) | $\widehat{\mathcal{L}}_{sup}(\Omega_e)$ | Time(s) | $\widehat{\mathcal{L}}_{sup}(\Omega_e)$ | Time(s) |
| $(24, 2)$ | 1.19 | 30 | 1.07 | 1.73 | 1.19 | 1.78 | 1.15 | 1.22 |
| $(40, 2)$ | 1.11 | 122 | 1.03 | 4.24 | 1.06 | 4.30 | 1.06 | 1.74 |
| $(56, 2)$ | 1.22 | 317 | 1.06 | 7.37 | 1.22 | 7.38 | 1.22 | 2.26 |
| $(24, 3)$ | 1.31 | 655 | 1.12 | 2.57 | 1.20 | 2.48 | 1.27 | 1.20 |
| $(40, 3)$ | 1.24 | 4,617 | 0.97 | 6.24 | 1.10 | 6.21 | 1.11 | 1.75 |
| $(56, 3)$ | 1.23 | 15,964 | 1.09 | 10.40 | 1.22 | 10.42 | 1.22 | 2.35 |

fatal loss estimate, but at the cost of significant computational effort. Random search, though less exhaustive, fails to provide sufficiently accurate estimates of fatal loss. Differently, sequential search yields better estimates under the same time consumption, closely approximating the ground truth in most cases. However, its time complexity increases significantly with the scale of QNN and the number of errors, making it impractical for large PQC implementations where errors are expected to increase. Our evolution-based low-complexity search strategy offers comparable estimates to sequential search but with approximately half the time cost for small-to-moderate QNN scales; this speedup even increases over 4 times on large QNN. Moreover, the time cost of our low-complexity search grows only marginally with the number of errors, e.g., only around $4\%$ from two errors to three errors, while this increment on sequential search is over $40\%$.

### 4.3 OVERVIEW OF EQUITABLE TRAINING

To solve the optimization problem presented in Eq. 5, we iteratively update the parameters $\boldsymbol{\theta}$ ithrough equitable training, as detailed in Algorithm 1. In each iteration, we perform signle-generation low-complexity search to estimate the supremum of $\mathcal{L}_{sup}(\Omega_e)$ for each data batch, so that error candidates evolve with the QNN training. This is followed by evaluating both the error-free model and the noisy model with identified fatal error $E^\star$. Crucially, we conduct multiple searches per iteration to frequently refresh the candidate set $\mathcal{S}_e$. This approach is essential for enhancing the

---

**Algorithm 1** Equitable noise-aware training on QNN

**Require:** error case set $\Omega_e$, QNN model $f$, training epoch $T$, scaling factor $\lambda$, learning rate $\eta$
**Ensure:** $\boldsymbol{\theta}^\star = \boldsymbol{\theta}^T$ for equitable QNN
1: initialize $\boldsymbol{\theta}^0$ and candidates $\mathcal{S}_e \subset \Omega_e$;
2: **for** $t = 1..T$ **do**
3:     **for** data batch $\boldsymbol{x}$ **do**
4:         $\boldsymbol{E}^\star, \{\mathcal{L}_{\boldsymbol{E}}\} \leftarrow \sup_{\boldsymbol{E} \in \mathcal{S}_e} \mathcal{L}(f(|\boldsymbol{x}\rangle, \boldsymbol{\theta}^{t-1}|\boldsymbol{E}), \boldsymbol{y})$
                 ▷ Deterministic noise model
5:         $\mathcal{S}_e \subset \Omega_e \leftarrow$ crossover&mutation$(\mathcal{S}_e, \{\mathcal{L}_{\boldsymbol{E}}\})$
6:         $\frac{\partial \mathcal{L}}{\partial \boldsymbol{\theta}}, \frac{\partial \mathcal{L}'}{\partial \boldsymbol{\theta}} \leftarrow \frac{\partial \mathcal{L}(f(|\boldsymbol{x}\rangle, \boldsymbol{\theta}^{t-1}), \boldsymbol{y})}{\partial \boldsymbol{\theta}^{t-1}}, \frac{\partial \mathcal{L}(f(|\boldsymbol{x}\rangle, \boldsymbol{\theta}^{t-1}|\boldsymbol{E}^\star), \boldsymbol{y})}{\partial \boldsymbol{\theta}^{t-1}}$
7:         $\boldsymbol{\theta}^t \leftarrow \boldsymbol{\theta}^{t-1} - \eta \frac{\partial \mathcal{L}}{\partial \boldsymbol{\theta}} - \eta \lambda \frac{\partial \mathcal{L}'}{\partial \boldsymbol{\theta}}$
8:     **end for**
9: **end for**

---

robustness of the QNN model against critical errors, given the large size of $\Omega_e$ and the necessity of covering fatal error cases for effective mitigation.

$\lambda$ **Selection.** $\lambda$ provides the trade-off between the normal accuracy and fatal accuracy of the QNN models. Although the determination of $\lambda$ can be evaluated by hyperparameter tuning, we share an empirical start as $\lambda = 0.5$. A discussion of the choice of $\lambda$ is presented in Appendix G.

## 5 EVALUATION

**Benchmark Strategies.** All QNN tasks are evaluated under the error-free training and the noise-injection training strategies, as benchmark methods. For the error-free strategy (baseline), we optimize QNN models with default architecture, excluding noise consideration during QNN training. For the noise injection strategy, we follow the assumption in Wang et al. (2022b) that one gate has the same and static rates to induce `Pauli-X`, `Pauli-Y`, and `Pauli-Z` error. For the injecting error rate, we

Table 2: The fatal accuracy (F.Acc) on QNN models (with 3/5/7/10 layers), trained by the baseline, noise-injection, and ours strategy. The evaluation tasks are MNIST (M2, M4, M10), FMNIST(F2, F4), and CIFAR(C2). All the models are trained on 50 epochs, with the same training configuration. Best F.Acc for each task is **highlighted**. Additionally, improvements of ours over the second-best are marked in green, while the difference to the best is in red when ours is not the best.

| | 3Layer | | | | | | 5Layer | | | | 7Layer | | | | 10Layer | | | |
|---|---|---|---|---|---|---|---|---|---|---|---|---|---|---|---|---|---|---|
| | M2 | M4 | M10 | F2 | F4 | C2 | M2 | M10 | F2 | F4 | M2 | M10 | F2 | C2 | M2 | F2 | F4 | C2 |
| Baseline | 0.303 | 0.023 | 0.067 | 0.247 | 0.099 | 0.380 | 0.152 | 0.025 | 0.163 | 0.059 | 0.146 | 0.011 | 0.166 | 0.378 | 0.141 | **0.195** | 0.056 | 0.341 |
| NI-L | 0.378 | 0.047 | 0.094 | 0.293 | 0.141 | 0.453 | 0.152 | 0.025 | 0.165 | 0.055 | 0.136 | 0.027 | 0.152 | 0.387 | 0.135 | 0.160 | 0.055 | 0.396 |
| NI-M | 0.273 | 0.033 | 0.062 | 0.333 | 0.147 | 0.444 | 0.148 | 0.023 | 0.164 | 0.041 | 0.149 | 0.030 | 0.167 | 0.388 | 0.127 | 0.164 | 0.039 | 0.367 |
| NI-H | 0.390 | 0.059 | 0.100 | 0.498 | 0.108 | 0.501 | 0.179 | 0.042 | 0.184 | 0.044 | 0.152 | **0.089** | 0.164 | 0.368 | 0.174 | 0.171 | 0.052 | **0.397** |
| **ours** | **0.564** | **0.136** | **0.139** | **0.673** | **0.272** | **0.536** | **0.184** | **0.074** | **0.211** | **0.064** | **0.191** | 0.086 | **0.198** | **0.407** | **0.183** | 0.173 | **0.057** | 0.390 |
| (+/−) | +0.174 | +0.077 | +0.039 | +0.175 | +0.126 | +0.036 | +0.004 | +0.032 | +0.027 | +0.005 | +0.039 | -0.003 | +0.031 | +0.019 | +0.009 | -0.022 | +0.001 | -0.007 |

generally categorize the error rates of a QNN model as "NI-L(ight)" ($p^X = p^Y = p^Z = 0.001$ for all qubits), "NI-M(oderate)" ($p = 0.005$), and "NI-H(eavy)" ($p = 0.01$).

**Metric.** We apply fatal accuracy (F.Acc) and noisy accuracy (N.Acc) to evaluate different NAT strategies. Specifically, fatal accuracy is derived from the lowest accuracy from error cases of $\Omega_e$ that contain high-probability errors. On the other hand, the noisy accuracy measures the model performance under a certain set of error rates on qubits. As the qubit error rate is unpredictable and dispersed (Section 3), we randomly pick three noisy environments based on the error rate levels. We denote $p_g = p_g^X = p_g^Y = p_g^Z$ on $g$-th gate, and each gate's error rate follows Gaussian distribution $p_g \sim \mathcal{N}(\mu_p, \sigma_p^2)$ s.t. $p_g > 0$ for a $G$-gate PQC. N.Acc(L) – the noisy accuracy under low error rates, $\mu_p = 0.001$ and $\sigma_p = 0.0005$; N.Acc(M) – for medium error rates $\mu_p = 0.005$ and $\sigma_p = 0.002$; N.Acc(H) – for high error rates $\mu_p = 0.01$ and $\sigma_p = 0.005$.

## 5.1 PROBLEM SOLVING: IMAGE CLASSIFICATION

The image classification task in QNN is in accordance with the definition used in classical computers. As image classification on QNN is still emerging, common QNN models can achieve good performance on easy datasets, such as MNIST Lecun et al. (1998), FMNIST Xiao et al. (2017), and CIFAR Krizhevsky et al. (2009). We evaluate the tasks MNIST2, MNIST4, MNIST10, FMNIST2, FMNIST4, and CIFAR2, which all have four-qubit input except MNIST10 with ten qubits. The details are provided in Appendix H.1.

**QNN Configuration.** W.L.O.G, we build the QNN with `U3` gates and `CU3` (Controlled-U3) gates. Each `U3` gate has three parameters indicating the three Euler angles to rotate a qubit on the Bloch sphere. We define one layer as "U3+CU3" layer, where each qubit has a `U3` gate followed by a `CU3` gate cyclically Wang et al. (2022a). We specify four QNN models, with 3 layers, 5 layers, 7 layers, and 10 layers. To apply our equitable training, we empirically assume $M = 1, 2, 2$, and 3 for each QNN model, representing larger PQC scales with more expected gate errors in practice.

### 5.1.1 FATAL ACCURACY EVALUATION

We demonstrate the fatal accuracies of different training strategies on our selected QNN models in Table 2. From an overall standpoint, our equitable training strategy consistently achieves the highest fatal accuracy across most tasks. It only slightly underperforms compared to the top-performing strategies on some large-scale tasks, such as the 7-layer MNIST10, where the difference is minimal. By comparing the noise injection (NI) strategy with the baseline method which does not account for noise, we observe a marked improvement in fatal accuracy, particularly in smaller QNNs (e.g., 3 layers). This underscores the importance of noise-adaptive training strategies in quantum execution. However, the NI strategy, which randomly samples error cases, faces challenges as the QNN scale increases due to the near-polynomial growth in the error case space. As a result, the advantage of the NI strategy diminishes for moderate to large QNNs (5 layers and above). This is evidenced by the NI strategy's marginal performance improvement over the baseline in fatal accuracy assessments. Only the "NI-Heavy" method, which has a higher likelihood of sampling errors, occasionally outperforms the baseline method with certain improvements.

On the other hand, our equitable training strategy employs the evolutionary search during model training, thereby assigning higher priority to locating fatal errors. In smaller QNN models, where the

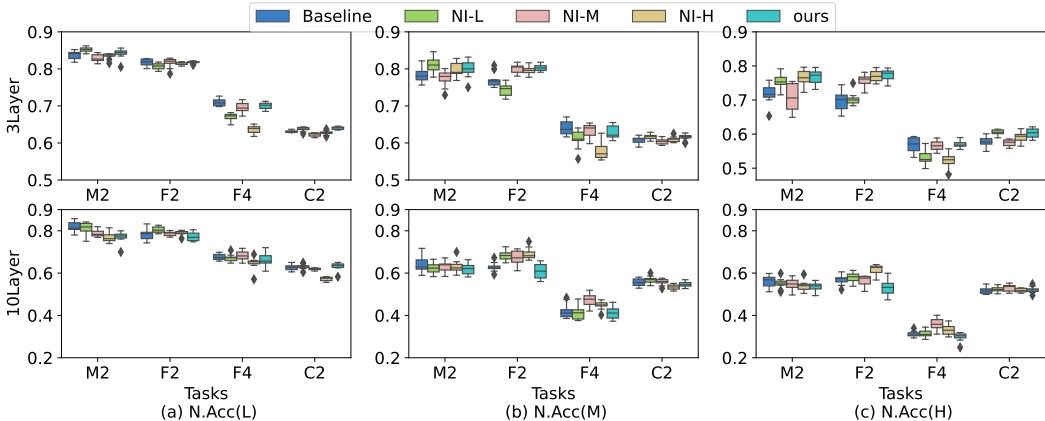

Figure 4: The noisy accuracy (N.Acc) of the 3/10-layer QNN model, optimized using various strategies under (L)ow, (M)edium, and (H)igh error rates. The boxplots summarize 10 runs for each task. Results for additional tasks are provided in Appendix I.2.

error set $\Omega_e$ is limited, our method significantly outperforms the Noise Injection (NI) strategy, for instance, achieving a 17.51% improvement in the FMNIST2 task for a 3-layer QNN. Due to more frequent access of fatal errors during training, our strategy also excels in moderate-scale QNN models, such as those with 5 layers. Nevertheless, as the model scale increases, our advantage diminishes. For example, our strategy ranks second best in the 7-layer MNIST10 and 10-layer FMNIST2 tasks, indicating reduced access to fatal errors and less optimization of the PQC parameters concerning these errors. Despite these challenges, our strategy still manages to deliver the best performance in over half of the large-scale QNN tasks.

### 5.1.2 NOISY ACCURACY EVALUATION

In Figure 4, we demonstrate the noisy accuracy of the small (3-layer) and large (10-layer) QNN models in different noisy environments. By incorporating additional parameters into PQC, the 10-layer model exhibits a higher likelihood of errors in noisy environments than 3 layers, leading to an obvious decline in noisy accuracy. Notably, additional layers would typically enhance performance without noise, creating antagonsim with the detrimental effects of more errors in the noisy environment. This is demonstrated by the evaluation for CIFAR2 task across 3-layer to 10-layer QNNs (shown in Figure 4 and 11), where the 5-layer QNN surpasses the 3-layer model under low error conditions (N.Acc(L)), yet accuracy decreases with further layer additions. In most cases, the adverse impact of errors outweighs the benefits of increased model complexity. Therefore, effective error mitigation strategies are essential for practical QML tasks.

Analysis of different strategies reveals that the baseline method excels only in simpler tasks, such as MNIST2, while strategies that incorporate noise considerations generally outperform the baseline in noisy environments across other tasks. However, this advantage is not markedly pronounced. Moreover, when compared to NI strategies, our equitable training achieves comparable noisy accuracy, leading in 8 out of 24 tasks. Combined with our method's superior performance in fatal accuracy, our equitable training not only matches the general performance of SOTA NAT strategies without compromise, but also significantly enhances the robustness of the QNN models against fatal errors during practical execution. Additionally, we evaluate these training methods on different tasks in real quantum environments with IBMQ backends, in Appendix H.1.1.

### 5.2 PROBLEM SOLVING: POS TAGGING

The sequential data modeling, such as natural language processing, is also receiving favors in the QML area Coecke et al. (2020). We take a case study of the part-of-speech (POS) tagging task to evaluate the NAT strategies in another QNN architecture, i.e., quantum long short-term memory (QLSTM) Chen et al. (2022). Detailed architecture is discussed in Appendix H.2. Given that current QLSTM models can only be performed on small-scale datasets Di Sipio et al. (2022), popular ones

(such as Penn Treebank Marcus et al. (1993)) are not practical for QLSTM application, and to our knowledge there is no corresponding work on them. Thus, we generate a small corpus that contains 20 sentences, whose vocabulary tags contain noun, verb, determiner, adjective, adverb, and pronoun.

**Results.** We demonstrate the experiment results in Table 5, evaluating the fatal accuracy and noisy accuracy on QNN models optimized with different strategies. For the fatal accuracy, our training strategy can achieve a significant improvement over baseline and noise-injection methods. Since the QNN scale and the cardinality of $\Omega_e$ are small and we have enough search times during the training iteration, the fatal loss is well addressed. This observation aligns with the results from the image classification problem. Besides, since the QLSTM architecture is

Figure 5: The fatal accuracy (F.Acc) and noisy accuracy(N.Acc) with different error rates, for the POS tagging task under different NAT strategies. The noisy accuracy is averaged on 10 runs.

|          | F.Acc | N.Acc(L) | N.Acc(M) | N.Acc(H) |
|----------|-------|----------|----------|----------|
| Baseline | 0.35  | $0.92^{\pm0.02}$ | $0.90^{\pm0.02}$ | $0.87^{\pm0.03}$ |
| NI-L     | 0.18  | $0.90^{\pm0.01}$ | $0.87^{\pm0.03}$ | $0.85^{\pm0.04}$ |
| NI-M     | 0.20  | $0.94^{\pm0.01}$ | $0.92^{\pm0.02}$ | $0.89^{\pm0.04}$ |
| NI-H     | 0.13  | $0.89^{\pm0.01}$ | $0.88^{\pm0.03}$ | $0.85^{\pm0.05}$ |
| **Ours** | **0.69** | $\mathbf{0.94}^{\pm0.01}$ | $\mathbf{0.93}^{\pm0.01}$ | $\mathbf{0.90}^{\pm0.04}$ |

a hybrid model, i.e., classical NN plus QNN, the parameters in the classical part can also help the robustness of QNN against fatal errors. In addition, our method can achieve comparable noisy accuracies with the noise-injection strategy under various error rates. This observation again proves the superiority of our method to provide robustness in fatal errors and dynamic noise environments.

### 5.3 PROBLEM SOLVING: REGRESSION

QNNs have been explored in regression tasks Wang (2017). Specifically, given a set of data samples $(\mathbf{X}, \mathbf{Y})$ following a mapping function $\tilde{f} : x \mapsto y$, QNN is trained to fitting this mapping by minimize the MSE loss function

$$\mathcal{L}(f(|\mathbf{X}\rangle, \boldsymbol{\theta}), \mathbf{Y}) = \sum_{x \in \mathbf{X}, y \in \mathbf{Y}} \|f(|x\rangle, \boldsymbol{\theta}) - y\|^2 \tag{6}$$

Similar to the image classification task, we build the QNN as a three-layer `U3+CU3` model with 2 qubits, which has in total $G = 12$ gates, thus the $\Omega_e$ collects all the error cases where $M = 1$. We evaluate the regression task on mapping function: $y = \sin(2x_1)\cos(x_2)$.

**Results.** Table 6 shows the fatal and noisy losses on training strategies evaluated. Our equitable training can achieve at least $57\%$ reduction on the fatal loss over other strategies. For noisy loss, our method cannot always outperform other NAT strategies on various error rates. Unlike the previous classification and tagging problem, the divergence of performance of different methods is significant. The baseline strategy without error consideration is prone to have lower loss in low error rate environment, while the noise injection with high-error-rate (NI-H)

Figure 6: The fatal MSE loss (F.Loss) and noisy MSE loss (N.Loss) with different error rates, for the regression task under different NAT strategies. The noisy loss is averaged on 10 runs.

|          | F.Loss | N.Loss(L) | N.Loss(M) | N.Loss(H) |
|----------|--------|-----------|-----------|-----------|
| Baseline | 1.958  | $\mathbf{0.008}^{\pm0.025}$ | $0.075^{\pm0.102}$ | $0.404^{\pm0.279}$ |
| NI-L     | 1.261  | $0.029^{\pm0.058}$ | $0.076^{\pm0.195}$ | $0.245^{\pm0.156}$ |
| NI-M     | 1.695  | $0.055^{\pm0.096}$ | $0.059^{\pm0.073}$ | $0.503^{\pm0.249}$ |
| NI-H     | 1.292  | $0.010^{\pm0.026}$ | $\mathbf{0.054}^{\pm0.070}$ | $0.196^{\pm0.172}$ |
| **Ours** | **0.543** | $0.048^{\pm0.019}$ | $0.123^{\pm0.083}$ | $\mathbf{0.134}^{\pm0.088}$ |

and our method perform better in high error rate environment. In addition, we notice that the noisy loss evaluation is not accurate, w.r.t. large standard deviation to the average N.Loss. This is also a motivation that we propose the F.Loss metric to evaluate the robustness of a model against error, which is error-rate independent.

## 6 CONCLUSION

We highlight the critical impact of dynamic noise and fatal errors on QNNs. Existing training and evaluation methods fall short facing a dynamically changing quantum noise environment. We propose fatal error analysis, which is independent of error rates and assesses QNN robustness against high-probability fatal errors. We also present the first noise-aware training strategy aimed at general fatal error mitigation, rather than a specific error rate. Our evaluations confirm the efficacy of our strategy for fatal errors; besides, QNNs optimized by our approach perform on par with SOTA NAT strategies under various error rates.

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

## A    PARAMETERIZED QUANTUM CIRCUIT OPTIMIZATION

Parameterized Quantum Circuit (PQC) involves a set of parameters $\boldsymbol{\theta}$ to convert the input state $|x\rangle$ to a new state $|\psi\rangle$ for measurement, denoted $f(|x\rangle, \boldsymbol{\theta})$ following Eq. 1. Optimization of $\boldsymbol{\theta}$ aims to minimize the loss function between the ground truth and the expectation value of an observable, $\mathcal{L}(f(|x\rangle, \boldsymbol{\theta}), y)$. During iterative updating of the parameter $\boldsymbol{\theta}$, two commonly adopted strategies are applied. The first scenario is for PQC optimization based on simulation, i.e., emulating the quantum propagation on the classical computer, so that the gradient $\partial \mathcal{L}/\partial \boldsymbol{\theta}$ is derived straightforwardly as normal NN training, then the parameter is updated through $\boldsymbol{\theta} \leftarrow \boldsymbol{\theta} - \eta \partial \mathcal{L}/\partial \boldsymbol{\theta}$.

Another scenario is to keep the quantum propagation on real quantum computer, and treat it as a black-box execution (due to the noise existence). Then, the gradient of $\boldsymbol{\theta}$ is estimated from the parameter-shift rule Simeone et al. (2022) as

$$\frac{\partial \mathcal{L}}{\partial \theta_g} \simeq \frac{\mathcal{L}(\boldsymbol{\theta} + \epsilon\theta_g|\mathcal{E}) - \mathcal{L}(\boldsymbol{\theta} - \epsilon\theta_g|\mathcal{E})}{2\epsilon}, \text{ where } \mathcal{E} \in \mathcal{D}[\mathcal{P}(p^X, p^Y, p^Z)]$$

where $\epsilon > 0$ is a small positive number and $\theta_g$ is the unit vector for the $g$-th parameter. $\mathcal{E}$ is sampled from the noise distribution $\mathcal{D}$ at the time of PQC execution on quantum computer. Afterwards, the parameter is updated iteratively. The benefit of this strategy is considering the real noise distribution $\mathcal{E}$ during PQC optimization, although it is unknown (as a black box). However, in practice, the noise distribution is changing dynamically and significantly, making it difficult for optimized PQC under one noise distribution to adapt to another, highlighting the necessity of our equitable training method.

## B    NOISE MITIGATION ON QNN MODELS

In addition to the NAT strategy that enhances the robustness of QNNs during training, various post-measurement and in-execution strategies have been developed to address quantum errors effectively. For instance, error extrapolation—executing the QNN multiple times under varying error conditions and synthesizing the results to approximate an error-free outcome—has been shown to reduce the impact of errors significantly Temme et al. (2017). Wang et al. propose an additional normalization of parametrized quantum circuit (PQC) measurements, leveraging the distribution of training data to refine measurement accuracy Wang et al. (2022b). Furthermore, Ravi et al. recommend employing dynamic decoupling, a method that counteracts decoherence and phase noise by strategically inserting multiple `Pauli-X` and `Pauli-Y` gates between the original gates of the circuit Ravi et al. (2022). These noise mitigation techniques, primarily aimed at enhancing QNN performance post-training, complement our training-centric strategy. Our NAT approach is designed to integrate seamlessly with these methods, ensuring robust performance throughout all phases of QNN operation—from training through execution.

In this study, we assess the integration of our NAT strategy with post-training noise mitigation techniques in a high-error-rate environment ($p = 0.01$), focusing on error mitigation performance. For the extrapolation method, we selected 10 random error rates between 0.001 and 0.05 during QNN inference and extrapolated to the error-free expectation value of the PQC measurement. In measurement nor-

Table 3: The N.Acc of 3-layer QNN under different noise mitigation strategies, under noisy environment of $p = 0.01$. Accuracies are collected from 10 runs. EX.—Extrapolation.

|  | M2 | M4 | F2 | F4 |
|---|---|---|---|---|
| Baseline | $0.718^{\pm 0.015}$ | $0.488^{\pm 0.023}$ | $0.710^{\pm 0.014}$ | $0.576^{\pm 0.010}$ |
| ours | $0.763^{\pm 0.009}$ | $0.520^{\pm 0.014}$ | $0.774^{\pm 0.007}$ | $0.562^{\pm 0.012}$ |
| ours+EX. | $0.811^{\pm 0.015}$ | $0.639^{\pm 0.008}$ | $0.802^{\pm 0.007}$ | $0.642^{\pm 0.010}$ |
| ours+Norm. | $0.766^{\pm 0.026}$ | $0.567^{\pm 0.022}$ | $0.772^{\pm 0.032}$ | $0.616^{\pm 0.022}$ |
| error-free | 0.856 | 0.691 | 0.822 | 0.715 |

malization, the QNN model was trained to standardize each qubit measurement across data batches. We conducted experiments using a 3-layer QNN on the MNIST2, MNIST4, FMNIST2, and FMNIST4 datasets, with results detailed in Table 3. While our training strategy significantly enhanced QNN model performance compared to the baseline, post-training noise mitigation further improved inference accuracy, approximating error-free conditions. Among the strategies, extrapolation yielded more substantial improvements due to multiple measurements, whereas measurement normalization offered enhancements with a single inference run.

## C    RELATED WORK

The noise-aware training (NAT) on QNNs has been recently proposed to apply robust training strategies during parameter optimization Wang et al. (2022b). Specifically, random noise injection is utilized during training, and other strategies, such as normalization and quantization, are applied as post-training robustness enhancement. Our work aligns with the in-training methodology, for example, noise injection, while it is orthogonal to the other strategies. The ensemble strategy is also proposed to improve performance with majority voting Qin et al. (2022); however, the multiplied model size will either burden the quantum resources or lengthen the execution time along the ensemble scale. Nevertheless, both works assume a static and unique noise distribution during QNN training, which contradicts reality, as quantum noise exhibits dynamic properties.

## D    DERIVATION OF PROBABILISTIC NOISE MODEL DERIVATIVE

Given a probabilistic noise model $f(|x\rangle, \boldsymbol{\theta})$ in Eq. 2, we derive $\partial f/\partial p_g^X$, where $p_g^X$ is the `Pauli-X` error rate on the $g$-th gate. We first rewrite the final density matrix during measurement as

$$\rho_G = \left( \prod_{i=g+1}^{G} \mathcal{E}_i \circ U_i(\theta_i) \right) (\rho_g),$$

$$\rho_g = \bar{p}_g I \rho'_{g-1} I + p_g^X X \rho'_{g-1} X + p_g^Y Y \rho'_{g-1} Y + p_g^Z Z, \tag{7}$$

$$\rho'_{g-1} = U_g(\theta_g) \left[ \left( \prod_{i=1}^{g-1} \mathcal{E}_i \circ U_i(\theta_i) \right) (\rho_0) \right] U_g^{\dagger}(\theta_g) \rho'_{g-1} Z$$

Therefore, $\partial f/\partial p_g^X = \partial f/\partial \rho_G \cdot \partial \rho_G/\partial p_g^X$.

We have $f = \text{Tr}[B\rho_G] = \sum_{i=1}^{2^Q} \sum_{j=1}^{2^Q} B_{i,j}(\rho_G)_{j,i}$ assuming a $Q$-qubit system, thus $\partial f/\partial \rho_G = B^T$ can be easily derived. For the second term, we further do the chain rule and calculate $\partial \rho_G/\partial p_g^X = \partial \rho_G/\partial \rho_g \cdot \partial \rho_g/\partial p_g^X$. We first represent a noisy gate operation as $\rho_m = \Lambda(\rho_{m-1})$, thus the derivative $\partial \rho_m/\partial \rho_{m-1}$ should be determined. For the first term in $\Lambda$, we have $\Lambda^1(\rho_{m-1}) = \bar{p}_m I U_m(\theta_m) \rho_{m-1} U_m^{\dagger}(\theta_m) I$; its derivative to $\rho_{m-1}$ can be written as

$$\frac{\partial \textbf{vec}(\Lambda^1)}{\partial \textbf{vec}(\rho_{m-1})} = \bar{p}_m (I^T U_m^*(\theta_m)) \otimes I U_m(\theta_m)$$

where $\textbf{vec}$ is the vectorization of a matrix, $U^*$ is the complex conjugate, and $\otimes$ is the Kronecker product. Similarly, we can derive the derivative in other three terms of $\Lambda$, and we denote $\Lambda'_m = \partial \rho_m/\partial \rho_{m-1} = \partial \Lambda/\partial \rho_{m-1} = (\partial \Lambda^1/\partial \rho_{m-1} + \partial \Lambda^2/\partial \rho_{m-1} + \partial \Lambda^3/\partial \rho_{m-1} + \partial \Lambda^4/\partial \rho_{m-1})$. The vector results can be converted to matrix format after calculation. Therefore, the derivative of $\partial \rho_G/\partial \rho_g$ can be calculated as $\prod_{i=G}^{g} \Lambda'_i$. For the term of $\partial \rho_g/\partial p_g^X$, it can be directly derived from Eq. 7 as

$$\frac{\partial \rho_g}{\partial p_g^X} = (X \rho'_{g-1} X - \rho'_{g-1})$$

To conclude all the derivations, we finally determine

$$\frac{\partial f}{\partial p_g^X} = \frac{\partial f}{\partial \rho_G} \cdot \frac{\partial \rho_G}{\partial \rho_g} \cdot \frac{\partial \rho_g}{\partial p_g^X} = \text{Tr}[\rho_{g^+}(X \rho'_{g-1} X - \rho'_{g-1})], \text{ where } \rho_{g^+} = B^T \prod_{i=G}^{g} \Lambda'_i$$

## E    HIGH-PROBABILITY ERROR CASE COLLECTION

While the motivation example in Figure 3 demonstrates the QNN performance under one-gate-error circumstance, the total error cases that a QNN model can incur is a combination problem. Upon $m$ errors occurring in an $G$-gate QNN, there are $\binom{G}{m}$ possible error cases. Evaluating every potential error case during NAT is impractical due to the extensive time required. Additionally, focusing on

---

**Algorithm 2** Error case set $\Omega_e$ collection

---

**Require:** The QNN architecture $(\{G_q\}, \{p_q\})$, the high-probability threshold $P_{th}$
**Ensure:** $\Omega_e$
1: $\Omega_e \leftarrow \emptyset, k = 1$;
2: $m \leftarrow \lceil \sum_{q=1}^{Q} G_q p_q \rceil$;
3: Collect $\Omega_{sub} = \{\boldsymbol{E}|\, |\boldsymbol{E}| = m\}$;                          ▷ Collect all the error cases with $m$ errors
4: $P \leftarrow \sum_{\boldsymbol{E} \in \Omega_{sub}} P(\boldsymbol{E})$
5: $\Omega_e \leftarrow \Omega_e \cup \Omega_{sub}$
6: **while** $P < P_{th}$ **do**
7:     $m_1 = m - k, m_2 = m + k$;
8:     **if** $m_1 \geq 0$ **then**        ▷ Collect the error cases in the direction of drecreasing the number of errors
9:         Collect $\Omega_{sub} = \{\boldsymbol{E}|\, |\boldsymbol{E}| = m_1\}$;
10:         $P \leftarrow P + \sum_{\boldsymbol{E} \in \Omega_{sub}} P(\boldsymbol{E})$;
11:         $\Omega_e \leftarrow \Omega_e \cup \Omega_{sub}$;
12:     **end if**
13:     **if** $m_2 \leq G$ and $P < P_{th}$ **then**        ▷ Collect the error cases in the direction of increasing the number of errors
14:         Collect $\Omega_{sub} = \{\boldsymbol{E}|\, |\boldsymbol{E}| = m_2\}$;
15:         $P \leftarrow P + \sum_{\boldsymbol{E} \in \Omega_{sub}} P(\boldsymbol{E})$;
16:         $\Omega_e \leftarrow \Omega_e \cup \Omega_{sub}$;
17:     **end if**
18:     $k \leftarrow k + 1$
19: **end while**

---

low-probability error cases is inefficient. Thus, it is necessary to define a collection that contains critical error cases for QNN performance. We define the "high-probability" error set as:

**Definition E.1** (High-probability error set $\Omega_e$). Given a QNN with $Q$ qubits, $G_q$ gates on each qubit, and logical error rate $p_q = p_q^X + p_q^Y + p_q^Z$ on each qubit, where $q \in [1, Q]$, $\Omega_e$ is defined as a set of error cases $\{\boldsymbol{E}\}$ who has the minimal cardinality, such that the total probility of all error cases (plus the error-free case if necessary) is larger than a threshold $P_{th}$ (e.g., 0.995). This definition is based on two assumptions:

① The error rates $\{p_q\}$ are independently distributed along the qubits, ignoring interference between the qubits. For example, crosstalk between qubits can be effectively mitigated by tuning the qubit frequency Ding et al. (2020). ② All the operations on a certain qubit have the same error rate distribution; this is reasonable because the execution of quantum gates is usually below the microsecond scale, e.g., only 103ns on `CNOT` gate in Noiri et al. (2022), which is short enough for a static qubit noise distribution. Note that this does not conflict with our previous observation on the large error rate variation on time scale, because other parts of circuit running on quantum computers, e.g., state profiling and repeated measurements, are the major time concern during PQC execution.

In order to collect $\Omega_e$, the error rate distribution should be analyzed. Considering the cases $\{\boldsymbol{E}\}$ where each qubit $q$ has $m_q$ gate errors, thus in total $m = \sum_{q=1}^{Q} m_q$ gate errors, the probability of this case group is

$$P(\{\boldsymbol{E}\}|\{m_q\}) = \prod_{q=1}^{Q} \binom{G_q}{m_q} (p_q)^{m_q} (1 - p_q)^{G_q - m_q}, \qquad (8)$$

where each qubit has an independent binomial distribution on the error occurrence. Likewise, the probability of error-free case is $P(\{\boldsymbol{e}\}|0) = \prod_{q=1}^{Q} (1 - p_q)^{G_q}$. Simultaneously, the expectation of the error number is $E(m) = \sum_{q=1}^{Q} G_q p_q$. Starting from $E(m)$, which is the most possible error case group, the error cases are collected in the increasing and decreasing directions of the error number. At the same time, $P(\{\boldsymbol{e}\}|\{m_q\})$ accumulates until $P_{th}$. We provide an algorithmic description in Algorithm 2. While our proposed error case evaluation should be independent of error rates, we do estimate the general error rate of a qubit $p_q$ during the collection of $\Omega_e$, as a summary of normal

quantum computer running. For example, $p_q = 0.001$ for `ibmq_algiers` and $p_q = 0.01$ for `ibmq_cusco`.

# F  Low-Complexity Search Design

To effectively estimate the supremum of losses, i.e., $\sup_{\boldsymbol{E} \in \Omega_e} \{ \mathcal{L}(f(|x\rangle, \boldsymbol{\theta}|\boldsymbol{E}), y) \}$, an efficient search strategy should be provided. Multiple in-training strategies are utilized in modern search tasks, such as differentiable search Liu et al. (2018), reinforcement learning Kaelbling et al. (1996), and evolutionary search Mirjalili & Mirjalili (2019).

## F.1  Search Strategy Selection

**Deficiency of Differentiable Search and Reinforcement Learning.** In the differentiable search strategy, all the candiate operations, such as $\{I, X, Y, Z\}$, should be differentiable during model execution. In this case, we use the probabilistic noise model (Eq. 2) to express each noisy gate operation, i.e., employing the density matrix representation of qubit states. Although this format is commonly used in theoretical analyses, such as noise channel modeling, it raises scalability concerns in classical computing. This is because the size of the density matrix is $2^Q \times 2^Q$ for a $Q$-qubit system; the computational complexity increases exponentially with the QNN scale, making it inefficient for practical use. For the reinforcement learning search, integration with the QNN model training involves setting up an agent model to decide the next action (error case selection) given the current error case and PQC parameters. However, the action space grows linearly with the number of QNN gates, which makes agent model training harder and more time-consuming. On the other hand, we utilize evolutionary algorithms, which avoid model training during the search, instead relying on multiple model inferences. Furthermore, as long as the population size per generation is appropriate, the search effort does not have to increase linearly with the QNN scale.

**Our Evolution-Based Low-Complexity Search Design.** We employ an evolutionary search strategy to minimize the online fatal loss in $\Omega_e$. Initially, we randomly generate $N_{pop}$ candidate error cases $\mathcal{S}_e$ for the first generation, evaluated against the deterministic noise model described in Eq. 3. For parent selection during crossover and mutation, we apply elite tournament selection within a substantial population. We then use uniform crossover and binary mutation techniques to generate the subsequent generation. It is important to note that the evolution of generations occurs concurrently with model training. Consequently, during each iteration of model training, only one generation is assessed based on the current state of the QNN model. The algorithm is referred to Algorithm 1.

**An Sequential Search Alternative.** Reviewing the composition of $\Omega_e$, it only contains error cases with certain error numbers, such as 1 and 2 when $M = 2$. Given this characteristic, we also propose another search strategy that approximates the supremum of $\Omega_e$, which is efficient for a quick search in small PQC scale. The algorithm of our sequential search strategy is based on the idea of greedy search, where the algorithm is provided in Algorithm 3. We categorize the error cases in $\Omega_e$ by the total number of errors $m = \sum_{q=1}^{Q} m_q$. From the group with the least number of errors, we evaluate all the cases, and find the error case with the largest loss. Then, this candidate is stored for further search. If we locate the worst error case with $m$ errors, then we boldly assume that these errors will also occur in fatal error cases with $(m+1)$ errors. Therefore, we proceed with the search for $(m+1)$ errors by fixing the $m$ errors as the worst error case of the $m$ errors, plus one error occurring in the rest of the $G - m$ gates, until reaching the largest number of errors in $\Omega_e$, denoted $M = \max\{m\}$. With this approach, we locate the fatal error case from the elements in $\Omega_e$. This strategy can only find an approximate supremem of losses with error, but will greatly reduce the searching time. Theoretically, the time complexity can be reduced from polynomial to linear on the circuit scale.

## F.2  Time Complexity Analysis

We assume that the evaluation time in a certain error case is a constant $C$, so the search time in $\Omega_e$ is dominated by the cardinality of $\Omega_e$. Denote the minimum number of errors in $\Omega_e$ as $m_0$, and the maximum number of errors as $M$. Thus, the cardinality of $\Omega_e$ is

$$|\Omega_e| = \sum_{m=m_0}^{M} \binom{G}{m} \tag{9}$$

---

**Algorithm 3** Low-complexity search to approximate the supremum of losses on selected error cases $\mathcal{L}_{sup}(\Omega_e)$

---

**Require:** error case set $\Omega_e$, loss function $\mathcal{L}(f(|x\rangle, \boldsymbol{\theta}|\boldsymbol{E}), y)$
**Ensure:** $\boldsymbol{E}^\star$ and approx $\widehat{\mathcal{L}}_{sup}(\Omega_e)$
   $[\{\boldsymbol{E}\}|_m] \leftarrow \Omega_e;$                                  ▷ group $\boldsymbol{E} \in \Omega_e$ by # of errors, $m$
   $t, M = \min\{m\}, \max\{m\};$
   $\Xi \leftarrow \emptyset;$
   **while** $t \leq M$ **do**
      extract group $\Gamma \leftarrow \{\boldsymbol{E}\}|_{m=t};$
      **if** $\Xi$ is empty **then**
         $\{\boldsymbol{E}\}_{sub} \leftarrow \Gamma;$
      **else**
         $\{\boldsymbol{E}\}_{sub} \leftarrow \{\boldsymbol{E}|\boldsymbol{E} \in \Gamma, \exists \xi \in \Xi|_{m=t-1}, \xi \subset \boldsymbol{E}\};$
      **end if**
      $\Delta \leftarrow \{y|y = \mathcal{L}(f(|\mathbf{X}\rangle, \boldsymbol{\theta}|\boldsymbol{E}), \mathbf{Y}), \boldsymbol{E} \in \{\boldsymbol{E}\}_{sub}\}$
      $\Xi \leftarrow \Xi \cup \{\max \Delta\}$
      $t \leftarrow t + 1$
   **end while**
   $\widehat{\mathcal{L}}_{sup}(\Omega_{\boldsymbol{E}}) \leftarrow \max \Xi$

---

where $G$ is the total number of gates in QNN. Due to the nature of the quantum error rates and the circuit scale, $M$ is usually less than 3 given a high probability threshold, e.g. $P_{th} = 0.995$. Thus, the growth of $\Omega_e$ size is polynomial along the circuit scale $G$. Therefore, the brute-force search in $\Omega_e$ costs time $C \cdot |\Omega_e| = O(G^M)$, which is roughly increasing with order $G^M$, given the fact that $G \gg M$ in practice. Here, we can conclude that the brute-force search time increases polynomially along the circuit scale, and exponentially along $M$ which is related to the threshold $P_{th}$ and the assumed general qubit error rate $p_q$.

For the **sequential search**, it only evaluates the $m$-error cases which contain the worst error case in $(m-1)$-error cases. Specifically, from $m_0$ to $M$ the total number of error cases that the low-complexity search evaluates is

$$N_{SS} = N_0 + \sum_{m=m_0+1}^{M} (G - m + 1) \tag{10}$$

where $N_0$ is the number of error cases with $m_0$ errors. Thus, the search time of the sequential search is linear to both the circuit scale $G$ and the $M$ (maximum of errors) in $\Omega_e$. Although the time complexity has been reduced from polynomial to linear, our experiments reveal that the sequential search still incurs significant time delays during large-scale QNN model training. This is attributable to the search procedure being repeated for each data batch and training iteration. For instance, a QNN model with parameters $(G, M) = (56, 3)$, assuming 50 training epochs and 100 data batches, would require approximately $8.32 \times 10^8$ model inferences. Such computational demands are impractical. Thus, a search strategy with even lower complexity is essential for efficient training of large-scale QNN models.

In our **evolution-based search**, the time complexity is primarily associated with the population size $N_{pop} = |\mathcal{S}_e|$ for each generation, compared to $N_{SS}$ in sequential search. While population selection is influenced by the QNN scale, our empirical evaluations indicate that $N_{pop}$ can increase sub-linearly relative to the QNN scale, particularly when there are sufficient data batches and training iterations, i.e., a large number of generations. For example, in our exploration, a QNN model with parameters $(G, M) = (56, 3)$ achieves satisfactory performance with $N_{pop} = 100$. Assuming 50 training epochs and 100 data batches, the QNN training requires only approximately $5 \times 10^5$ model inferences. This approach is substantially more efficient than the sequential search strategy. Thus, our evolution-based low-complexity strategy can well address the scalability issue during the fatal loss estimation.

## G   ABLATION STUDY ON $\lambda$

In the optimization problem (Equation 5), $\lambda$ tunes the updating focus between the error-free model and the fatal-error model. In Figure 7, we demonstrate the $\lambda$ tuning on the fatal accuray and error-free accuracy using our 3-layer QNN example. The best fatal accuracy occurs at $\lambda = 0.6$. As the observation, we find that the ratio of fatal loss $\mathcal{L}(f(|\mathbf{X}\rangle, \boldsymbol{\theta}|\boldsymbol{E}^\star), \mathbf{Y})$ over error-free loss $\mathcal{L}(f(|\mathbf{X}\rangle, \boldsymbol{\theta}), \mathbf{Y})$ is usually in the range of $[1.5, 6]$ in all of our experiments, which means that these two losses are on the same scale yet fatal loss could be 2 times larger. Thus, $\lambda$ around 0.5 is an appropriate choice to slightly emphasize the fatal loss while still taking into account the error-free loss, which is adopted in our experiments.

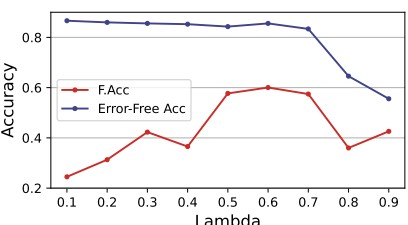

Figure 7: The fatal accuracy and error-free accuracy of the 3-layer QNN on MNIST2 task, with varing $\lambda$.

## H    EVALUATION DETAIL

### H.1    IMAGE CLASSIFICATION

For image classification, an input image $x$ is encoded by rotation gates Wang et al. (2022a), which is represented by a qubit state vector $|x\rangle$. Later, the PQC is conducted for the vector porcessing and the final state is measured through a certain basis, e.g., the computational basis. The loss is evaluated as

$$\mathcal{L}(f(|\mathbf{X}\rangle, \boldsymbol{\theta}), \mathbf{Y}) = \text{CrossEntropy}(f(|\mathbf{X}\rangle, \boldsymbol{\theta}), \mathbf{Y})$$

where $(\mathbf{X}, \mathbf{Y})$ is the image-label pair of training dataset. In Table 4, we demonstrate the dataset information for the evaluated image classification tasks. For QNN models optimized on two-class tasks, the PQC measurements of qubit 0 and qubit 1 are summed up as the score for the first class, while the summation of the other two measurements is for the second class. For other tasks, each qubit measurement represents the score for one class.

Table 4: The information of tasks selected to evaluated NAT strategies on image classification.

| Task | Class | Input Size | # of qubits | Encoder |
|---|---|---|---|---|
| MNIST2 | $(3, 6)$ | $4 \times 4$ | 4 | $R_y$-$R_z$-$R_x$-$R_y$ |
| MNIST4 | $(0, 1, 2, 3)$ | $4 \times 4$ | 4 | $R_y$-$R_z$-$R_x$-$R_y$ |
| MNIST10 | $(0, ..., 9)$ | $10 \times 10$ | 10 | $R_y$-$R_z$-...-$R_y$ |
| FMNIST2 | (Dress, Shirt) | $4 \times 4$ | 4 | $R_y$-$R_z$-$R_x$-$R_y$ |
| FMNIST4 | (T-Shirt, Trouser, Pullover, Dress) | $4 \times 4$ | 4 | $R_y$-$R_z$-$R_x$-$R_y$ |
| CIFAR2 | (Auto, Bird) | $4 \times 4$ | 4 | $R_y$-$R_z$-$R_x$-$R_y$ |

We further provide an illustration of 3-layer QNN with 4 qubits, in Figure 8. In this PQC, we have 12 `U3` gates and 12 `CU3` gates, i.e., $G = 24$. However, please note that these general gates are not naturally supported in most quantum computers. Instead, they must be decomposed to the basis gates belonging to the targeted quantum computers. An example of gate decomposition is illustrated in Figure 9. Here, the 24-gate PQC is decomposed to 204 basis gates for quantum execution. With respect to architecture and operation planning, further optimization is usually performed, thus the final gate number is smaller than the initial results.

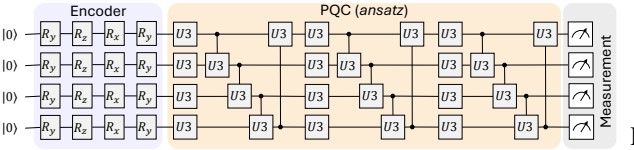

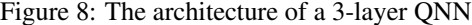

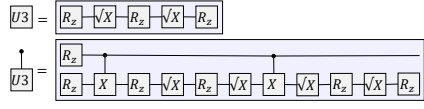

Figure 8: The architecture of a 3-layer QNN

Figure 9: The decomposition of `U3` and `CU3` gates, following primitives `CNOT`, `RZ`, `SX`, and `X` (IBM basis gates).

In our evaluation, we specify four QNN models, with 3/5/7/10 "`U3`+`CU3`" layers.    For MNIST2/MNIST4/FMNIST2/FMNIST4/CIFAR2 tasks, these model have 24/40/56/80 gates respectively; yet for the MNIST10 task with 10 qubits, these models have 60/100/140/200 gates. For

the training configuration, we universally setup 50 epochs for PQC optimization, where the fatal loss is considered starting from the 10th epoch. We employ the Adam optimizer (`lr = 0.03` and `wd=1e-4`) with cosine annealing learning rate decay.

### H.1.1 Evaluation on Real Quantum Computers

While we manually configure the noise environments based on the calibration data of known quantum computers, we hereby further deploy our trained QNN models on actual quantum computers to assess their performance in realistic noisy environments. Specifically, we conduct a series of selected tasks on the `ibmq_brisbane` quantum computer, through the qiskit toolkit, including M2/M4/F2/F4 tasks on 3/5-layer QNN models. The execution time on real quantum computers significantly exceeds that of classical simulations due to the measurements being derived from multiple circuit execu-

Table 5: The inference accuracy of M2/M4/F2/F4 tasks on 3/5-layer QNN models, evaluted on `ibmq_brisbane` environment. Accuracy is derived from 500 test samples, where 8192 shots are conducted for each sample's observation.

| | 3Layer | | | | 5Layer | | | |
|---|---|---|---|---|---|---|---|---|
| | M2 | M4 | F2 | F4 | M2 | M4 | F2 | F4 |
| baseline | 0.872 | 0.714 | 0.842 | 0.750 | 0.874 | 0.734 | 0.838 | 0.712 |
| NI-L | 0.870 | 0.652 | 0.826 | 0.680 | 0.860 | 0.730 | 0.858 | 0.742 |
| NI-M | 0.868 | 0.650 | 0.834 | 0.710 | 0.876 | 0.712 | 0.824 | 0.736 |
| NI-H | 0.860 | 0.604 | 0.832 | 0.650 | 0.880 | 0.694 | 0.834 | 0.730 |
| ours | 0.876 | 0.694 | 0.830 | 0.712 | 0.872 | – | 0.850 | 0.760 |

tions (shots); therefore, we limit our testing to 500 samples per task, with each sample measurement comprising 8192 shots. The accuracy results are presented in Table 5. The results presented largely corroborate our observations in Sec. 5.1.2, demonstrating that our method can achieve QNN performance comparable to both the baseline and the noise injection strategy. Notably, our method generally improves inference accuracy across most tasks over noise injection (NI) strategy, yet sometimes underperform the baseline method. Since our evaluations are conducted on a SOTA IBM quantum computer, that has relatively low error rates during execution, the observed improvements are not significant. This is further evidenced by the inference accuracy in Table 5, which closely approximates the results from error-free evaluations (shown in Table 3). In this scenario, the baseline model can sometimes achieve better error-free accuracy due to noise absence during training, yet its performance deteriorates when errors occur, as we highlight in Table 2.

### H.2 POS Tagging

Quantum long short-term memory (QLSTM) use PQC to represent the classical LSTM counterpart, which has the following mathematical construction:

$$
\begin{aligned}
g_t &= \sigma(f_1(|v_t\rangle, \boldsymbol{\theta}_1)) \\
i_t &= \sigma(f_2(|v_t\rangle, \boldsymbol{\theta}_2)) \\
\tilde{C}_t &= \tanh(f_3(|v_t\rangle, \boldsymbol{\theta}_3)) \\
c_t &= g_t \times c_{t-1} + i_t \times \tilde{C}_t \\
o_t &= \sigma(f_4(|v_t\rangle, \boldsymbol{\theta}_4)) \\
h_t &= f_5(|o_t\tanh(c_t)\rangle, \boldsymbol{\theta}_5) \\
y_t &= f_6(|o_t\tanh(c_t)\rangle, \boldsymbol{\theta}_6)
\end{aligned}
\tag{11}
$$

where $(v_t, g_t, i_t, c_t, o_t, h_t, y_t)$ are the default LSTM variables Yu et al. (2019), yet we use $g_t$ to represent the forget gate (rather than $f$) to avoid confuse with our QNN measurement notation. $\Omega_e$ has $M = 1$ to satisfy $P_{th}$.

## I Supplementary Figures

### I.1 Qubit Error Rates on Other Quantum Computers

As the complementary material, we further collect the `Pauli-X` gate error rate on other IBM quantum computers, i.e., `ibmq_mumbai`, `ibmq_cario`, `ibmq_brisbane`, and `ibmq_sherbrooke`. The qubit error landscape is shown in Figure 10.

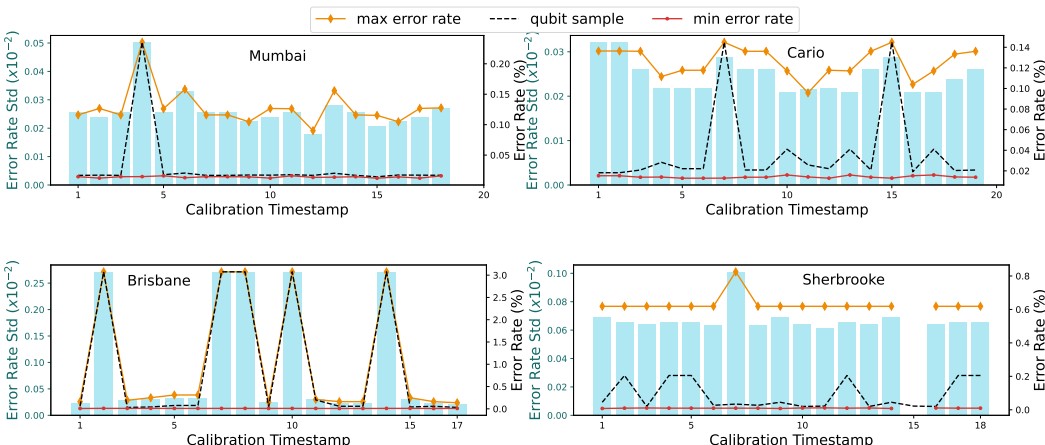

Figure 10: The demonstration of large variation of qubit error rates on both time scale and qubit scale. We record the `Pauli-X` gate error rates of all qubits of four IBM quantum computers, `ibmq_mumbai`, `ibmq_cario`, `ibmq_brisbane`, and `ibmq_sherbrooke`. We sample 20 calibration data in a consecutive 72-hour frame. The barplot (left y-axis) demonstrates the standard deviation of error rates on qubits of each calibration, along the time scale. The line plot (right y-aixs) shows the maximum/minimium qubit error rate of all qubits for each calibration, and the dash line highlights the qubit error rate track which has the largest variation along the calibration time window.

## I.2 NOISY ACCURACY OF OTHER TASKS

As the complementary of the N.Acc results in Figure 4, we visualize other tasks in Figure 11. Just as shown in Figure 4, we highlight analogous findings here that our equitable training approach attains accuracy similar to other methods. This underscores our progress towards achieving a QNN model with high N.Acc and high F.Acc, indicating that the model's performance remains stable across different errors, even in fatal cases.

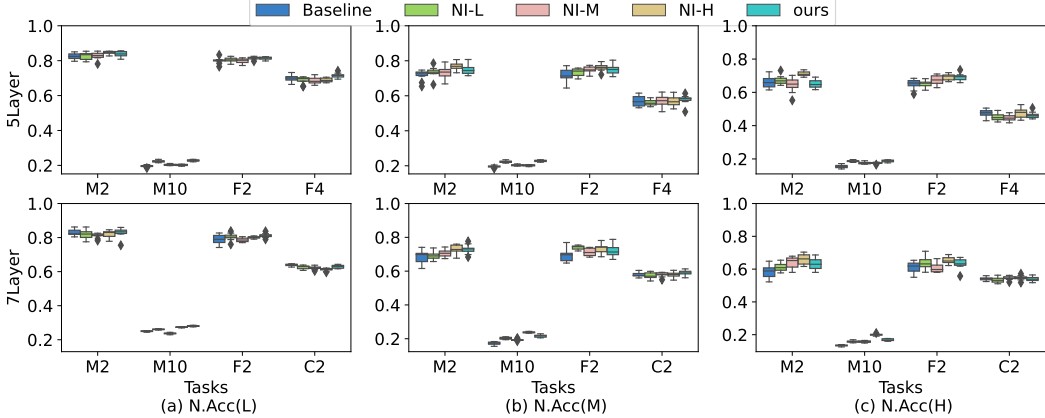

Figure 11: The noisy accuracy (N.Acc) of the 5/7-layer QNN model, optimized using various strategies under (L)ow, (M)edium, and (H)igh error rates. The boxplots summarize 10 runs for each task.

