# OpenReview forum: "Robust Quantum Neural Networks Against Dynamic Noise Landscape in the NISQ Era"
_ICLR.cc/2025/Conference — ICLR 2025 Conference Withdrawn Submission_

### Official Review · Reviewer_5oG4 · 2024-11-01

**Soundness:** 2
**Presentation:** 3
**Contribution:** 2
**Rating:** 3
**Confidence:** 4

**Summary:**

The authors address the problem of quantum neural networks (QNNs) facing dynamic and unpredictable noise in the current noisy intermediate-scale quantum (NISQ) era and propose a novel NAT strategy that adapts to both standard and severe "fatal error" conditions, coupled with a low-complexity search method for efficiently identifying these fatal errors during the optimization process. They introduce a metric called “Fatal Accuracy” for evaluating the specific impact of `fatal errors', those that severely degrade QNN performance, underscoring the inadequacies of existing NAT approaches that do not address such error sensitivity.

**Strengths:**

- The methods introduced in this work are simple and concise.
- Numerical results show the efficiency of low-complexity searching and training strategy with dynamical noise.

**Weaknesses:**

- Based on Eqn. (5), Equitable Training can be interpreted as standard QNN training with a modified loss function, where the loss function combines a weighted sum of the noiseless loss and the worst-case noisy loss (fatal loss). Essentially, this strategy seeks a parameter set that mitigates fatal noise while preserving accuracy in the noiseless scenario. In the beginning, the authors present the noise variance in the calibration of real devices, which raises a natural question: why not define the loss function as a weighted sum of loss functions from randomly sampled noise scenarios? Or take the weighted sum between noiseless loss and the average noisy loss instead? (As the evaluation or real environment is unpredictable)

- In the Evaluation section, the authors benchmark the performance of different strategies; however, I noticed that Figure 4 does not clearly support the superior performance of the equitable training strategy. For instance, in some cases, the NI-H approach outperforms equitable training in the N.Acc (H) setting. This raises the question of whether the fatal loss function is perhaps not well-suited for scenarios with random noise.

- In Tables 5 and 6, I noticed a significant gap between the fatal and noisy accuracy (or loss) in the baseline case, even under heavy noise conditions. Is the fatal case truly this severe, or could this discrepancy results from choice of noise parameters? Also, your equitable training strategy shows no obvious advantage over noise injection strategy in the N.Acc senario.

- The authors conclude that their noise-aware training strategy is "aimed at general fatal error mitigation, rather than a specific error rate." However, from another perspective, the fatal error model identified through the low-complexity search is itself a specific error model with a fixed error rate for each gate. In particular, I am curious how the fatal errors are determined in Table 2; specifically, what is the error set used in evaluation? Was it derived from a Gaussian distribution? Additionally, it would be beneficial to include comparisons on real devices or noise data directly from device calibration.

Minor issues:
- In the section 4.3, there are typos “ithrough” and “signle-generation” at the third and fourth row.
- The authors use an error-free strategy as the baseline; however, it’s important to note that this baseline is error-free during training but includes errors during evaluation. Therefore, I suggest a more careful treatment of the term "error-free" to avoid potential confusion. Instead, the author can consider use “error-neglect” or just baseline to replace it.
- Table 5, 6 are labeled with Figure 5, 6.

Overall, while the paper addresses an important challenge of fatal error in quantum machine learning, its contributions feel somewhat limited and may lack broad applicability, particularly in dynamically noisy environments. The proposed equitable training approach has potential, but there are aspects of the methodology and evaluation that remain unclear. It would be better if the authors can address the questions and concerns raised, particularly around the choice of error models and the empirical validation on real devices.

**Questions:**

See my comments above.

---

### Official Review · Reviewer_PKbK · 2024-11-02

**Soundness:** 3
**Presentation:** 2
**Contribution:** 3
**Rating:** 6
**Confidence:** 4

**Summary:**

The paper presents a novel noise-aware training (NAT) for quantum neural networks (QNN) that is able to dynamically identify and mitigate quantum noise that arises in NISQ devices. The authors make comparisons with state of the art NAT approaches. The errors. overall cost when present may not perfectly match their probability of occurrence.

**Strengths:**

The novel NAT method improves the performance of the tested QNNs over different benchmarks.

**Weaknesses:**

The authors focus only over a certain subset or errors without discussing how the method can be applied alongside error correction methods  during execution (combining this NAT strategy with error mitigation is discussed in the appendix). The efficiency of the combined strategies and identifying the highest contributors of fatal errors would be of interest for the community.

I am not completely sure whether bolding of certain phrases as well as some other choices to emphasise complies with the style guide.

**Questions:**

The authors suggest that their proposed NAT method is highly efficient during training and it would be interesting for the reader to see a comparison to SOTA NAT evaluations required for different cases as an addition to the accuracy of performance visualisations.

Would the dynamical nature of the suggested method allow for training QNN models via classical simulation and then run experiments with the trained models over different QPUs (superconducting, trapped ions, neutral atoms)?

---

### Official Review · Reviewer_hehS · 2024-11-02

**Soundness:** 2
**Presentation:** 3
**Contribution:** 2
**Rating:** 3
**Confidence:** 3

**Summary:**

In this paper, the authors propose a noise-aware training (NAT) method for quantum neural networks (QNN) that addresses dynamic noise in NISQ devices. The method makes the QNN more robust to errors, especially fatal errors that make QNN performance worse than random guessing.

The authors first show the impact of dynamic noise and fatal errors on QNN. The authors then propose noise-aware training by optimizing the "noisy accuracy" so that QNN performs well under average error rates, and the "fatal accuracy" so that QNN performs well under fatal errors. The fatal error set is searched using a genetic algorithm. Finally, the authors empirically evaluate their approach to various QNN tasks, such as image classification, POS tagging, and regression. The result shows that the proposed method outperforms existing static noise injection techniques in handling fatal error conditions while maintaining competitive performance under average error rates scenario.

**Strengths:**

- The proposed approach, which introduces a fatal error term in the loss function, is simple and novel. Moreover, the authors provide a scalable to compute this term with a low-complexity search.
- The experimental evaluation of the paper is comprehensive with various problems ranging from image classification, NLP task, and regression problems.
- Overall, the writing is clear and the presentation is well structured.

**Weaknesses:**

- The experiments in the paper are all classically simulated with noise based on calibration data from known quantum computers. I think the biggest flaw of the paper is that when the authors tested their method on a real quantum device, it did not give any significant improvement, in fact, it was worse than the baseline error-free strategy almost half the time. I think this result should be in the main text with a strong justification and explanation of why the proposed method is still important, but the authors put it in Appendix H.1.1. This gives the impression that the authors are trying to sweep this result under the rug.
- Furthermore, the authors mention in the same appendix that "Since our evaluations are performed on a SOTA IBM quantum computer, which has relatively low error rates during execution, the observed improvements are not significant", which shows the weakness of the proposed method. As quantum hardware gets better and better and error rates get lower and lower, the fatal error may become less and less likely and the proposed method will become irrelevant. It is also undeniable that at some point, the NISQ era will come to an end and possibly the early fault-tolerance era will come.
- As shown in Table 2, the effectiveness of the proposed method deteriorates as the number of layers in the QNN increases. Meanwhile, from Figure 4, it seems that the noisy accuracy still increases as the number of layers increases.  I believe that solving complex tasks with QNN could also extend to 10 or more layers, making the proposed method less effective.
- In Table 2, the results for some tasks are also not shown (e.g., M4 in 5, 7, and 10Layer, F4 in 7Layer), which might raise some suspicion about the result. If the reason is scalability, why is there a result of F4 in 10Layer? Please justify or provide the full experiments in the appendix.

Conclusion: Based on the comment above, the weaknesses outweigh the strengths. Therefore, I believe that this paper is not good enough to be accepted at the ICLR.


Minor issues:
- Please use the same x-axis range in Figure 4.
- In Figure 5, to be fair, the 0.94 +- 0.01 of the NI-M in the N.Acc(L) should also be bold.
- Typo “minimium” in Figure 2 caption line 175.

**Questions:**

Please address the weaknesses I mentioned above. I would also be interested in how the number of qubits instead of the number of layers affects the results because the image classification only shows results for 4 and 10 qubits.

---

### Official Review · Reviewer_awQv · 2024-11-05

**Soundness:** 3
**Presentation:** 2
**Contribution:** 2
**Rating:** 5
**Confidence:** 3

**Summary:**

The paper tackles the challenge of training QNNs in noisy intermediate-scale quantum (NISQ) environments. The authors propose a new NAT strategy focused on "fatal errors"—specific noise instances that significantly degrade QNN performance. Using a low-complexity search method to identify these critical errors during training, the approach enhances QNN robustness under dynamic noise. Case studies like image classification and regression confirm improved resilience.

**Strengths:**

This paper identifies a limitation in existing noise-aware training (NAT) approaches, which assume static error rates. Instead, the paper introduces a novel NAT approach that adapts to dynamic noise fluctuations and fatal errors, advancing over traditional methods that overlook error variability across time and qubits. The low-complexity search algorithm efficiently identifies fatal errors, optimizing robustness without extensive computation.

**Weaknesses:**

- Scalability: The article only presents 4-qubit QNN experiments, which is insufficient to effectively demonstrate the scalability of the proposed method. Current classical quantum simulators can provide simulations with at least 20+ qubits, the authors should consider conducting experiments on a larger scale.

- Typo: Line 352 'ithrough' and Line 354 'signle'.

For more weakness, please refer to the questions.

**Questions:**

1. About scope of the work: What's the difference between noise mitigation techniques (Acharya et al. (2023)) and the proposed method？

2. Real quantum noise: Has the experiment considered running on real, diverse quantum computers to validate the training of this dynamic noise-aware method?

3. The experiment seems to consider only three types of noise: bit flip errors (Pauli-X), phase flip errors (Pauli-Z), and bit-phase flip errors (Pauli-Y). However, can the proposed model effectively generalize to a broader range of noise models, such as incoherent errors.

---

### Note · Authors · 2024-11-19

I have read and agree with the venue's withdrawal policy on behalf of myself and my co-authors.